# Detection Analysis and Study of Genomic Region Variability of JCPyV, BKPyV, MCPyV, HPyV6, HPyV7 and QPyV in the Urine and Plasma of HIV-1-Infected Patients

**DOI:** 10.3390/v14112544

**Published:** 2022-11-17

**Authors:** Sara Passerini, Carla Prezioso, Annalisa Prota, Giulia Babini, Luigi Coppola, Alessandra Lodi, Anna Chiara Epifani, Loredana Sarmati, Massimo Andreoni, Ugo Moens, Valeria Pietropaolo, Marco Ciotti

**Affiliations:** 1Department of Public Health and Infectious Diseases, “Sapienza” University of Rome, 00185 Rome, Italy; 2IRCSS San Raffaele Roma, Microbiology of Chronic Neuro-Degenerative Pathologies, 00163 Rome, Italy; 3Infectious Diseases Clinic, Polyclinic Tor Vergata, Viale Oxford 81, 00133 Rome, Italy; 4Department of Medical Biology, Faculty of Health Sciences, University of Tromsø—The Arctic University of Norway, 9037 Tromsø, Norway; 5Virology Unit, Polyclinic Tor Vergata, Viale Oxford 81, 00133 Rome, Italy

**Keywords:** HPyVs, HIV-1, immunosuppression, genomic region variability, HPyVs-related diseases

## Abstract

Since it was clearly established that HIV/AIDS predisposes to the infection, persistence or reactivation of latent viruses, the prevalence of human polyomaviruses (HPyVs) among HIV-1-infected patients and a possible correlation between HPyVs and HIV sero-status were investigated. PCR was performed to detect and quantify JCPyV, BKPyV, MCPyV, HPyV6, HPyV7 and QPyV DNA in the urine and plasma samples of 103 HIV-1-infected patients. Subsequently, NCCR, VP1 and MCPyV LT sequences were examined. In addition, for MCPyV, the expression of transcripts for the LT gene was investigated. JCPyV, BKPyV and MCPyV’s presence was reported, whereas HPyV6, HPyV7 and QPyV were not detected in any sample. Co-infection patterns of JCPyV, BKPyV and MCPyV were found. Archetype-like NCCRs were observed with some point mutations in plasma samples positive for JCPyV and BKPyV. The VP1 region was found to be highly conserved among these subjects. LT did not show mutations causing stop codons, and LT transcripts were expressed in MCPyV positive samples. A significant correlation between HPyVs’ detection and a low level of CD4+ was reported. In conclusion, HPyV6, HPyV7 and QPyV seem to not have a clinical relevance in HIV-1 patients, whereas further studies are warranted to define the clinical importance of JCPyV, BKPyV and MCPyV DNA detection in these subjects.

## 1. Introduction

Human polyomaviruses (HPyVs) are small non-enveloped DNA viruses widely distributed among the population [1]. To date, 15 HPyVs have been discovered [2,3,4,5,6,7,8,9,10,11,12,13,14,15]. After primary infection that usually occurs in childhood, HPyVs establish a subclinical and persistent infection in healthy subjects. However, since immune depression can lead to viral reactivation, HPyVs can be pathogenic in hosts with weakened immunity [16]. The first two human HPyVs identified and described in the literature were BKPyV and JCPyV [2,3]. BKPyV is associated with nephropathy in renal transplant patients and hemorrhagic cystitis in hematological patients, respectively, whereas JCPyV is the etiological agent of progressive multifocal leukoencephalopathy (PML), a demyelinating disease of the central nervous system (CNS). PML could occur after viral reactivation in oligodendrocytes and frequently is associated with an impairment of the immune system [17,18,19]. Other HPyVs can be pathogenic in immunocompromised patients. Merkel cell polyomavirus (MCPyV) is the causative agent of the Merkel cell carcinoma (MCC), a rare and aggressive neuroendocrine skin cancer [6]. Infection sustained by MCPyV and immunosuppression are strongly correlated. In fact, it has been reported that individuals whose immune system is chronically suppressed, such as HIV patients, are more at risk of developing MCC than immunocompetent subjects [20]. Trichodisplasia spinulosa polyomavirus (TSPyV) has been isolated for the first time in the plucked facial spines of a heart transplant patient with trichodisplasia spinulosa, which has been reported as the possible cause of this rare skin disease in immunocompromised individuals [8]. HPyV6 and HPyV7 were originally identified in non-diseased skin. Then, further studies demonstrated their association with pruritic dermatoses in immunocompromised patients, whereas their role in the skin diseases of immunocompetent patients is still unclear [21]. To date, other HPyVs have not been definitely associated with specific human diseases. However, although KIPyV and WUPyV have been frequently isolated from the respiratory specimens of subjects with acute respiratory symptoms and in COVID-19 patients, in which they may behave as opportunistic respiratory pathogens [22], no evidence for an association with respiratory disease has been reported [23]. QPyV is the most recently discovered HPyV [15], and it was isolated for the first time in the feces of an 85-year-old man. Recently, although it was found in the urine of individuals affected by systemic lupus erythematosus (SLE) and multiple sclerosis (MS) patients [24], its DNA was not detected in urine and plasma samples from HIV-1 positive patients [24].

Regarding genome, HPyVs share a structure that, functionally, could be divided into three regions: early, late and non-coding control region (NCCR). The early region encodes for the regulatory proteins involved in viral transcription and replication, such as large T antigen (LTAg) and small t antigen (stAg) [1,25]. Among them, LTAg seems to have oncogenic properties due to its capability to induce transformation in host cells. The late region encodes for the structural viral capsid proteins VP1 and VP2 [1]. Some HPyVs express additional regulatory and capsid proteins [25]. Interspersed between the early and late regions is the NCCR, which encompasses the origin of replication (ORI) and the regulatory sequences for early and late genes’ expression [25].

The coding regions are strongly conserved compared to NCCR showing extreme variability. Specifically, based on NCCR variations, JCPyV and BKPyV strains were classified as an “archetype” or “prototype” [26]. An archetype is considered the transmissible form of the virus among the population and is usually detected in healthy subjects, whereas a prototype is the rearranged form, frequently associated with HPyVs’ related disease. Indeed, rearrangements in the NCCR sequence influence HPyVs’ tropism and may affect viral replication as well as HPyVs’ pathogenic properties [17]. As reported in previous studies, rearranged variants have been associated with human diseases such as BKPyV-associated nephropathy or the progressive multifocal leukoencephalopathy (PML) caused by JCPyV [17,27]. Moreover, nucleotide variations and insertions were also described in the MCPyV NCCR region. Although it has been demonstrated that these mutations produced an increase in viral replication, the impact on the pathogenic features of the virus remains to be determined [28]. As previously reported, the deficit of the immune system can be linked to a genetic defect, to an iatrogenic condition (i.e., transplantation, chemotherapy) or to a viral infection or reactivation, as in the case of human immunodeficiency virus (HIV)-1 infection [29,30]. In this latter condition, the decrease in CD4+ can favor persistence or reactivation of HPyVs, influencing clinical outcomes of HIV/AIDS patients. JCPyV, BKPyV and MCPyV detection in HIV-1-infected individuals has been widely described in the literature, whereas little information is available about HPyV6, HPyV7 and QPyV prevalence in these patients [24,31].

HPyV6 and HPyV7 are natural inhabitants of the healthy skin virome [21]. However, they have also been detected in other specimens taken from immunocompromised individuals, including urine from HIV-1-infected patients. Limited data are available about QPyV’s genoprevalence and seroprevalence; therefore, it is still unclear what could be the cell tropism of this virus. QPyV showed a high degree of similarity to HPyV6 and HPyV7 at the nucleotide level, suggesting its DNA could be detected in clinical samples where HPyV6 and HPyV7 have been reported.

Based on this background, in this study, the prevalence of JCPyV, BKPyV, MCPyV, HPyV6, HPyV7 and QPyV in the urine and plasma samples of HIV-1-infected patients was evaluated. Moreover, in order to establish a possible association between HPyVs and HIV sero-status, the detection of HPyVs DNA was correlated to the level of CD4+, whereas the HPyV load was correlated with age, gender and HIV-1 load at enrollment.

## 2. Materials and Methods

### 2.1. Patients and Samples

In total, 103 paired plasma and urine samples were collected from HIV-1-infected individuals (70 males and 33 females; mean age: 48.8 years; median age: 49) from December 2021 to July 2022 at the Infectious Diseases Clinic of the Polyclinic Tor Vergata, Rome, Italy. The study was carried out according to the Declaration of Helsinki, and approval was granted by the Ethics Committee of the Policlinico Tor Vergata (protocol number: 0027234/2018, 19 December 2018). All patients gave written informed consent. Demographic and clinical characteristic are presented in Table 1.

### 2.2. DNA Extraction

Total DNA was extracted from urine and plasma samples using a Quick-DNA MiniPrep (Zymo Research, Irvine, CA, USA), in accordance with the instructions of the manufacturer. The extracted nucleic acids were eluted in a final volume of 50 μL, and DNA was evaluated for its PCR suitability by amplifying the β-globin gene sequences [32].

### 2.3. Screening of JCPyV, BKPyV, MCPyV, HPyV6 and HPyV7 DNA by Real-Time PCR

Quantitative real-time PCR (qPCR) was carried out to detect and quantify JCPyV, BKPyV, MCPyV, HPyV6 and HPyV7 DNA using a 7300 real-time PCR system (Applied Biosystems, Waltham, MA, USA) following published protocols. Each sample was analyzed in triplicate, and viral loads (given as the mean of at least three positive reactions) were expressed as genome equivalents (gEq)/mL. Negative and positive controls were included in each qPCR session. Standard curves were obtained from serial dilutions (range: 10^5^–10^2^ gEq/mL) of plasmids containing, respectively, the entire JCPyV, BKPyV, MCPyV, HPyV6 and HPyV7 genomes. The lower detection limit of the qPCR system was 10 DNA copies of the target gene per amplification reaction, corresponding to 10 genome equivalents per reaction (10 gEq/reaction). Specifically, the presence and quantity of JCPyV DNA was detected by a qPCR system able to detect a 54 bp amplicon in JCPyV LTAg region [33,34,35]. Quantitative determination of BKPyV DNA was carried out by the Thermo Scientific AcroMetrix BKPyV Panel containing intact, encapsidated viral particles (VP1) [34,36]. MCPyV DNA quantity was analyzed, using primers and probes for MCPyV sT, as previously described [37]. For HPyV6 and HPyV7, qPCRs with primers targeting the VP1 gene were performed [38]. The employed primers are listed in Table 2.

### 2.4. Reverse Transcription and PCR

Total RNA was extracted from the MCPyV positive specimens in order to study the expression of transcripts from the MCPyV LT gene. Total RNA was extracted with the Quick-RNA Miniprep Plus Kit (Zymo Research) and treated with DNase to avoid the amplification of viral DNA. The RNA was reverse-transcribed in cDNA with ZymoScript RT PreMix Kit (Zymo Research) including all the necessary components needed to perform robust reverse transcription. After the RNA sample is added to ZymoScript RT PreMix, the reaction is incubated for 2 min at 25 °C to initiate the reverse-transcription step. After the reverse-transcription step, the extension phase occurred at 25 °C for 10 min. After inactivation of the RT enzyme at 95 °C for 1 min, an aliquot of the reverse transcription reaction mixture (1 μL) was used for the subsequent PCR amplification carried out with primer sequences to determine the *LT* gene expression (LT-RNA-F sequence (5′→3′): GATCAGGAGGATTCAGCTTCG, nucleotide position based on MCC350 genome: 910–930; LT-RNA-R sequence (5′→3′): CAGAGGATGAGGTGGGTTCC, nucleotide position based on MCC350 genome: 1133–1152; predicted product size: 242 bp) [39]. The β-globin gene was amplified to confirm the presence of PCR-amplifiable cDNA.

### 2.5. Detection of QPyV DNA by Qualitative PCR

The presence of QPyV DNA was investigated by qualitative PCR, using primers able to amplify a 254 bp fragment of the VP1 gene: 5′-CAAAGTACAACACCACTTGTAG-3′ (nucleotides in 1973–1994; BK010702) and 5′-TTCTGAGGTTTCAGGAATTGCC-3′ (nucleotides 2205–2226; BK010702) [24]. The complete QPyV VP1 sequence was synthesized and cloned in the HindIII/XhoI sites of plasmid pcDNA3.1+C-HA by GenScript (BioPartner, Leiden, The Netherlands) and used as a positive control. Amplification products were analyzed by electrophoresis in 2% agarose gels stained with gel red and observed under UV.

### 2.6. Sequencing of the NCCR, VP1 and MCPyV LTAg Regions

HPyVs DNA positive samples were subsequently amplified for NCCR, VP1 and MCPyV *LTAg* following published protocols [6,40,41,42,43,44]. The employed primers are listed in Table 3. PCR products were analyzed on 2% agarose gels with gel red under UV light. The amplified products were purified using miPCR purification kit (Metabion, Plannegg, Germany) and sequenced in a dedicated facility (Bio-Fab research, Rome, Italy). All obtained BKPyV, JCPyV and MCPyV sequences were compared to their reference strains deposited in GenBank (BKPyV strain WW: AB211371; JCPyV: NC_001699 for Mad-1 or AB038249 for CY; MCPyV strain MCC350: EU375803). Sequence alignment was performed by ClustalW2 on the European Molecular Biology Laboratory–European Bioinformatics Institute (EMBL–EBI) website using default parameters [45]. Sequences obtained from amplification of the VP1 gene typing region of BKPyV and JCPyV were analyzed on the basis of single nucleotide polymorphisms (SNPs) used to classify the BKPyV subtypes/subgroups [17] and the JCPyV genotypes/subtypes [46]. MCPyV *LTAg* sequences were compared with the GenBenk reference sequences MCC350 and EU375803.

### 2.7. Statistical Analysis

HPyVs detection were summarized by counts and proportions. If continuous variables were normally distributed, they were expressed as mean ± SD; if not, they were expressed by median and range. The χ^2^ test was performed to evaluate differences in the viral detection, and the Mann–Whitney U-test for non-normally distributed continuous variables was applied to analyze differences between patients. A *p* value < 0.05 was considered statistically significant.

## 3. Results

### 3.1. Detection of JCPyV, BKPyV, MCPyV, HPyV6, HPyV7 and QPyV

Fifty-seven urine (57/103, 55%) and sixteen (16/103, 15%) plasma samples tested positive for JCPyV. Fourteen patients that tested positive for JCPyV in their urine samples also tested positive in their plasma specimens. Twenty-six urine (26/103, 25%) and four (4/103, 3.8%) plasma samples tested positive for BKPyV. BKPyV DNA was concomitantly detected in the urine and plasma samples of two patients. MCPyV was detected in 22 out of 103 urine samples (21%) and in 8 out of 103 plasma samples (7.7%). Among patients who tested positive for MCPyV DNA, two patients presented viral DNA in both their urine and plasma samples.

HPyV6, HPyV7 and QPyV were not detected in any sample. qPCR showed a JCPyV load mean value of 6 × 10^7^ gEq/mL in urine and of 6 × 10^5^ gEq/mL in plasma. For BKPyV, the mean viral load was 1 × 10^5^ gEq/mL in urine and 2.5 × 10^3^ gEq/mL in plasma. Finally, the MCPyV mean viral load was 5 × 10^3^ gEq/mL in urine and 1 × 10^3^ gEq/mL in plasma (Table 4).

### 3.2. HPyVs Co-Infections Analysis

Examining the HPyVs co-infection patterns, the most frequent combination was JCPyV, BKPyV and MCPyV, found in 12 (11.6%) urine samples from HIV-1-infected individuals. All co-infections are summarized in Table 5.

The detection of BKPyV, JCPyV and MCPyV in all positive analyzed samples showed a significant correlation with a low level of CD4+, ~200 CD4+/mm^3^ (Table 6), whereas no significant association was found between the presence of HPyVs DNA *versus* age, gender and HIV-1 load at enrollment.

### 3.3. Analysis of NCCR Structure

Sequencing analysis showed, as expected, an archetype NCCR structure in all analyzed urine samples positive to JCPyV and BKPyV DNA detection and a high degree of homology with MCC350; the EU375803 strain was in all MCPyV DNA positive urine and plasma samples (Table 7). An archetype NCCR organization, with the occurrence of some point mutations, was observed in all JCPyV (Table 8) and BKPyV plasma samples (Table 9). Specifically, regarding the JCPyV NCCR structure organization, the T to G transversion at nucleotide position 37 in box B within the Spi-B binding site; the G to A transition at nucleotide position 108 in box C; and the G to A transition at nucleotide position 217 in box F within the NF-1 binding site were observed (Table 8). The nucleotide numbering refers to the NCCR sequence of the non-pathogenic JCPyV CY strain (deposited sequence in GeneBank: JCPyV AB081613) [46]. Analyzing the BKPyV NCCR, A to G transitions at nucleotide position 5 in box R and position 19 in box P were revealed (Table 9). The nucleotide numbering refers to the BKPyV archetypal NCCR organization (deposited sequence in GeneBank: BKPyV AB263926) [47].

### 3.4. Analysis of JCPyV, BKPyV and MCPyV VP1

An additional specific PCR was undertaken to detect the presence of the *VP1* region in all BKPyV, JCPyV and MCPyV positive samples. Regarding BKPyV, the 327 bp *VP1* gene sequence was amplified in all urine samples and in four plasma samples. The amplified PCR products were then sequenced in order to classify each BKPyV strain into the corresponding subtype/subgroup, analyzing the SNPs within the amplified *VP1* region and aligning our 327 bp typing isolates with the consensus sequences generated for each BKPyV subtype/subgroup [48]. Based on the SNPs and on the consensus sequences, subtype I/subgroup b-2 was detected in all urine samples (26/26) and in two plasma samples (2/4), while subtype II was detected in the remaining two plasma samples (2/4).

The SNPs were also used to determine the JCPyV genotype of our positive samples. In particular, among the 57 positive urine, a prevalence of the European genotype 1A was observed (57/57). In the plasma samples, a prevalence of the European genotype 1A was observed in 9/16 samples, whereas European genotype 1B was detected in 7/16 samples. It is noteworthy that, in two JCPyV positive plasma samples, two JCPyV VP1 sequence showed the S269F and S267L point mutations within the VP1 receptor-binding region.

The amplified VP1 MCPyV DNA fragments, spanning from nucleotide position 3156 to 4427, were compared with the reference sequence of the prototype strain MCC350: EU375803 [6]. VP1 from one urine sample (1/22) displayed the 4289 T to A transversion, which resulted in Thr47Ser amino acid substitution located in the VP1 N-terminus; whereas, the VP1 of one plasma sample (1/8) contained the 4222 T to A transversion, which resulted in Ser251Phe amino acid substitution in the apical loop. VP1 sequencing analysis performed on the remaining MCPyV positive samples showed some nucleotides’ differences, with respect to the reference strains that did not produce any amino acid change in the derived protein sequence.

### 3.5. Detection of MCPyV LTAg by PCR, Analysis of the LTAg Trascripts and DNA Sequencing Analysis of the Full-Length LTAg

The LT1 and LT3 primers produced amplicons with sizes of 440 and 309 bp, respectively. Of the 22 positive urine samples, MCPyV DNA was detected in 17/22 samples with the LT1 primers and in 22/22 samples with the LT3 primers. Among the eight positive plasma samples, MCPyV DNA was detected in 6/8 samples with the LT1 primers and in 8/8 samples with the LT3 primers. The expression of the MCPyV LTAg transcripts (nt positions 910–1152, corresponding to exon 2) was examined at the RNA level by RT-PCR. In total, 10/22 positive urine samples and 2/8 positive plasma samples expressed the LTAg transcript. The specific amplification of the LTAg transcripts was confirmed by *in-service* sequencing. The sequence analysis of the full-length LTAg at nt positions 151–3102 was performed, comparing our trascripts with the wild-type non-tumor-derived MCPyV strain JN038578. Although several non-synonymous mutations were detected throughout the sequence, included at the C terminus of LTAg, no mutations causing stop codons were observed.

## 4. Discussion

HPyVs include ubiquitous, clinically silent viral pathogens that establish a symbiotic relationship with their human hosts [49]. Diseases associated with HPyVs have been described in immunocompromised individuals, such as HIV-1/AIDS patients, or patients with immunological aberrations. Current evidence indicates that HPyV-specific T cells and also neutralizing antibodies play a crucial role in the control of HPyVs replication and recovery from HPyV-associated diseases [18,19,30,50,51,52].

BKPyV is the causative agent of BKPyV-associated nephropathy in kidney transplant recipients and hemorrhagic cystitis in bone marrow transplant patients. Primary sites for BKPyV replication are the renal epithelium and uroepithelium, resulting in the lytic destruction of these cells. BKPyV reactivation and disease have been observed in other immunocompromised conditions, such as systemic lupus erythematosus in other solid organ transplant recipients, and in patients with HIV/AIDS [53]. Regarding JCPyV, the etiological agent of PML, immune-altering conditions, in which cases of PML have been reported, include lymphoproliferative diseases such as lymphomas and leukemias, myeloproliferative diseases, transplantation, chemotherapy, multiple sclerosis (MS) and inherited immunodeficiences [54,55,56]. The most common underlying cause of immunosuppression leading to JCPyV reactivation is the HIV/AIDS condition, in which a lytic infection of oligodendrocytes in the brain could develop in PML.

MCPyV has been implicated in the etiology of MCC, a rare but aggressive form of skin cancer [6]: *Feng* et al. identified sequences corresponding to MCPyV in 8 of 10 MCC tumors with viral DNA that, in 6 of the 8 MCC-positive tumors, showed a clonal integration pattern. Interestingly, the integrated form of MCPyV in tumors is predicted to encode a truncated LTAg, due to mutations within the second exon of the LTAg gene. This truncation would result in the loss of *LTAg* domains required for viral DNA replication and p53 binding but does not affect the domains required for inducing cell cycle progression, suggesting that this predicted protein may retain its transformation capability [53]. The primary risk factors for MCC development include, ultraviolet (UV) light exposure, advanced age and immunosuppression [49]. Although it is well-defined that HIV/AIDS predisposes to viral infection and to development of MCC, up to now, very few studies have focused on the MCPyV prevalence and viral load in HIV-1-positive individuals without MCC [57]. Published data demonstrated that the levels of anti-MCPyV IgG in HIV/AIDS patients were significantly higher than those in non-AIDS HIV-infected patients, and the prevalence of MCPyV-DNA in the peripheral blood mononuclear cells (PBMCs) of HIV/AIDS and non-AIDS HIV-infected patients was 17% and 16%, respectively [58]. Moreover, another study by *Fukumoto and colleagues* found that 9/23 (39%) serum samples from HIV patients, without highly active antiretroviral therapy (HAART) therapy, were MCPyV-positive [59]. Lastly, MCPyV DNA was found in 2/19 (11%) urine samples of HIV patients, as reported by *Torres and colleagues* [60], and in 10/66 (15%) urine samples, 7/66 (10%) plasma samples and 23/66 (35%) rectal samples, as reported by *Prezioso and colleagues* [28].

Serological studies have demonstrated that HPyV6 and HPyV7 infections are ubiquitous in the healthy adult human population. The presence of HPyV6 and HPyV7 DNA has been examined in different biological samples of healthy controls and different patient groups, in a quest to determine the cell tropism and the possible association of these viruses with diseases. Although clear associations with diseases have not been established, except in severely immunocompromised HIV/AIDS patients, in which HPyV6 and HPyV7 can cause pruritic dermatoses, these two HPyV are considered natural inhabitants of the healthy skin virome dermatoses [31].

The genoprevalence and seroprevalence of QPyV have, so far, not been well studied, so it remains to be determined whether this is truly an HPyV. A study conducted by *Prezioso and colleagues* showed that QPyV sequences could be detected in the urine of systemic lupus erythematosus (SLE) patients, multiple sclerosis (MS) patients and pregnant women but not in HIV-positive patients [24]. This result suggests that QPyV viruria may not be uncommon in individuals with a compromised immune system (SLE and MS patients) or with a unique immune condition occurring during pregnancy. This prompted us to further investigate whether QPyV DNA could be detected in clinical samples of HIV-positive subjects.

In this framework, the aim of our study was to evaluate the prevalence of JCPyV, BKPyV, MCPyV, HPyV6, HPyV7 and QPyV in the urine and plasma samples of HIV-1-infected patients in order to better understand the HPyVs tissue tropism and to provide new insights into the possible pathogenic role of these viruses in human diseases. Moreover, in order to establish a possible association between HPyVs and HIV sero-status, the detection of HPyVs DNA was correlated to the level of CD4+, whereas the HPyV load was correlated with age, gender and HIV-1 load at enrollment.

Our results showed that JCPyV, BKPyV and MCPyV DNA were detected in urine and plasma samples, while none of the samples contained HPyV6, HPyV7 or QPyV DNA. According to the measured viral load, JCPyV replicated at a higher level either in the urine (10^7^ gEq/mL) or plasma (10^5^ gEq/mL) compartments, followed by BKPyV (10^5^ gEq/mL and 10^3^ gEq/mL, respectively) and MCPyV (10^3^ gEq/mL, in both compartments). These results confirm that the renal epithelium and blood are the preferred targets of BKPyV and JCPyV infection and replication but less preferential infection sites for MCPyV [61].

It is reasonable to speculate that the renal epithelium and blood may represent latent/persistent sites for MCPyV rather than sites of active replication. It is well-recognized that primary HPyVs infection is followed by the establishment of an asymptomatic latency state, possibly in the lymphoid, neuronal, kidney and hematopoietic tissues characterized by low-level replication and excretion, for example, in urine [61].

HPyV6 and HPyV7 that are commonly detected in normal skin [7] were not detected in the urine and plasma samples of these patients. This finding partly contrasts with another study performed by our group that detected HPyV6 in 9.1% of the urine specimens from HIV-1 patients [31]. The presence of HPyV7 in plasma samples from two immunocompromised transplant patients was reported by Ho et al. and associated to pruritic rash [62]. However, the plasma samples were weakly positive for HPyV7 (10^3^ copies/cell) compared to the viral load detected in the skin biopsies (10^3^ copies/cell and 10^2^ copies/cell). To our knowledge, no authors have reported the presence of HPyV7 in blood samples of HIV-1 patients. Therefore, whether the HPyV6 and HPyV7 viruria is sporadic or linked to a particular clinical condition remains to be determined.

QPyV was already investigated by our group in a series of immunocompromised patients, including HIV-1 positive patients, but none of the urine and plasma samples from the HIV-1 patients revealed the presence of the virus by real-time PCR [24], as reported in the present study, suggesting that the virus could not be present in these sample types. As already reported, the genoprevalence of QPyV appears to be low [24], so additional studies, directed at investigating the presence of QPyV in other body compartments, are required to understand the role, if any, of QPyV in human diseases.

Regarding the HPyVs co-infections analysis, JCPyV–BKPyV–MCPyV were the HPyVs most frequently detected, and the higher detection rate could be explained by the wide circulation of these polyomaviruses among the population [63].

A significant correlation was found between the low level of the CD4+ cell count and the detection of BKPyV, JCPyV and MCPyV in all the analyzed samples. This result could suggest that the immunological alterations induced by the HIV/AIDS state might facilitate an HPyVs infection and reactivation.

In the context of HIV infection, it has been described that the detection of BKPyV and JCPyV DNA in urine increases concomitantly with a decrease in CD4+ cell counts [63,64]; although, the importance of other factors during JCPyV reactivation, when the viruria does not correlate with the degree of immunosuppression, has also been reported [65].

Among the novel HPyVs investigated in this study, MCPyV is the only one that is detected in HIV-1 patients and shows a significant correlation with a low level of CD4+ cell count. Considering the oncogenic potential of MCPyV, it could be worth monitoring this virus in patients with a weakened immune system.

Since it is well-known that the NCCR is a hypervariable region that may determine viral replication efficiency, and cellular tropism and NCCR rearrangements have been associated with specific human diseases’ development, such as nephropathy for BKPyV and PML for JCPyV [17,27], in this study, we characterized, in positive samples, HPyVs NCCR sequences. Sequence analysis of NCCR of BKPyV, JCPyV and MCPyV revealed an archetype structure in all the urine analyzed, confirming that rearrangements are uncommon in this anatomical site.

In the positive JCPyV and BKPyV plasma samples, instead, some point mutations were observed. Analysis of JCPyV’s NCCR reported two-point mutations that have been previously described in the literature [66,67]: the 37T-to-G nucleotide transversion in box B and the 217G-to-A nucleotide transition in box F. The 37T-to-G nucleotide transversion falls in box B within the binding site for Spi-B, a transcriptional factor that is involved in viral replication and neurovirulence, since, when it binds to box B, it is able to activate JCPyV promoter in both glial and non-glial cells [68]. The nucleotide transition 217 G to T locates in box F, within the NF-1 transcriptional factor binding site. NF-1 has been shown to increase the expression of JCPyV’s early and late genes in glial cells, which are permissive to viral replication [69,70]. Therefore, these point mutations, as previously reported, may precede NCCR reorganization, leading to PML-associated variants [71]. Furthermore, a recent wide analysis on the NCCR of JCPyV strains isolated from patient samples, including PML-type strains’ samples, indicates that the Spi-B site might enhance cellular tropism from the rearrangements in NCCR [72].

Mutations affecting these binding sites in the NCCR of BKPyV are able to affect gene expression [73]. However, the nucleotide changes observed in the plasma samples that were positive for BKPyV in this study did not involve any transcriptional factor binding sites.

In order to study the properties of the JCPyV, BKPyV and MCPyV strains circulating in HIV-1-infected patients, the obtained VP1 sequences were compared with JCPyV, BKPyV and MCPyV reference strains. The JCPyV, BKPyV and MCPyV VP1 regions were revealed to be highly conserved with nucleotide variations that did not determine amino acid changes.

The molecular analysis of the BKPyV VP1 coding sequence between nucleotides 1744 and 1812 (amino acids 61 to 83) allows for the definition of BKPyV genotypes and their different distribution in human populations [74]. In particular, subtype I (further divided into four subgroups, each of which has a unique geographical distribution pattern: I/a, I/b-1, I/b-2 and I/c) is widespread throughout the world; subtype IV (further divided into six subgroups with their own geographical distribution pattern: IV/a-1, IV/a-2, IV/b-1, IV/b-2, IV/c-1 and IV/c-2) is prevalent in East Asia and part of Europe; subtypes II and III are rarely detected throughout the world [75]. Furthermore, all BKPyV subtypes/subgroups are defined by the presence of specific nucleotide substitutions and deletions within the NCCR sequence [76]. We found European subtype I/b-2 and subtype II. In fact, alignment of all NCCR sequences, isolated from each BKPyV-positive clinical sample of our patients, with the prototypic NCCR sequence proposed by Yogo and colleagues [77], revealed that the point mutations detected in these sequences were typical of the BKPyV subtype I/subgroup b-2 and subtype II.

In two HIV-positive patients in which JCPyV DNA was detected, point mutations were found in the VP1 coding region (S269F in case 3 and S267 L in case 4). These mutations occurred in VP1 sites for sialic acid binding, changing the virus binding properties to its receptor or driving it to alternative receptor usage. It has been speculated that mutations within this site may alter the preference of JCPyV capsids from sialylated glycans outside the CNS to non-sialylated glycans inside the CNS, which might explain the infection and replication in glial cells [78].

The predominant genotypes among JCPyV positive samples were 1A and 1B, confirming that our results are in agreement with the genotypes commonly detected in the European population [79,80].

MCPyV, in contrast with other PyVs such as SV40, BKPyV and JCPyV, which employ sialic acid for cell attachment and internalization [81,82], utilizes sulfated carbohydrates as attachment receptors and sialic acids in the post-attachment process [83]. Specifically, MCPyV, for cell attachment and internalization, recognizes the N-acetyl neuraminic acid (Neu5Ac) binding site motif, located in the apical loop [84]. In this study, the Ser251Phe change within the first neutralizing epitope loop of VP1 was observed [85]. Since no analysis to investigate MCPyV immune escape was performed, it is solely possible suppose that this mutation could concur, to limit the antibody response against the MCPyV that is typically generated versus the epitopes exposed at the surface of VP1 [86].

Considering the widespread prevalence of MCPyV across the body and the proposed role of MCPyV in tumors other than MCC, in this study we investigated the presence of the MCPyV LTAg at the DNA and RNA levels. LT1 and LT3 primer sets, which are commonly used to detect MCPyV by PCR, were employed and were able to detect LTAg DNA in all the positive samples. In good concordance with previous reports [87,88], the LT3 primers showed the highest sensitivity with respect to LT1.

Overall, 10 out of 22 positive urine samples and 2 out of 8 positive plasma samples had detectable levels of LTAg transcripts. These results could be explained by assuming that the quantity and quality of the extracted nucleic acid are critical for reliably detecting RNA transcripts. The full-length LTAg sequence analysis showed that, although several non-synonymous mutations were detected throughout the sequence, including at the C terminus of LTAg, no stop codons were generated. So the truncated LTAg that represents the molecular signature of MCPyV in MCCs was not observed in this context.

In conclusion, this study adds important information about HPyVs prevalence, viral load and genomic region behavior in HIV-positive individuals and suggests that further studies are warranted to define whether HPyV6, HPyV7 and QPyV have a clinical relevance in these subjects.

## Figures and Tables

**Table 1 viruses-14-02544-t001:** Demographic and clinical characteristic of HIV-infected patients.

*Features*	*Population*	
** *Patients* **	103	
** *Sex, n (%)* **	M	F
	70 (68%)	33 (32%)
** *Mean age, years (SD)* **	48.8 (±11.9)	
** *Median age, years* **	49	
** *Mean HIV-RNA (cp/mL)* **	4.67 × 10^5^	
** *CD4+ cells/mm^3^, n (%)* **		
*≤200*	68/103 (66%)	
*>200*	35/103 (34%)	
** *Stadium (CDC Atlanta)* **		
*A1*	15/103 (14.6%)	
*A2*	15/103 (14.6%)	
*A3*	10/103 (9.7%)	
*B1*	2/103 (1.9%)	
*B2*	13/103 (12.6%)	
*B3*	16/103 (15.5%)	
*C2*	5/103 (4.9%)	
*C3*	27/103 (26.2%)	
** *Treatment n (%)* **		
*Naive*	7/103 (6.8%)	
*Experienced*	96/103 (93.2%)	
** *AIDS-defining illnesses, n (%)* **	29/103 (28%)	
*Wasting syndrome*	4/103 (3.9%)	
*Neurotoxoplasmosis*	1/103 (0.9%)	
*Pneumocystis jirovecii pneumonia*	10/103 (9.7%)	
*Visceral Kaposi’s sarcoma*	1/103 (0.9%)	
*Kaposi’s sarcoma*	2/103 (1.9%)	
*Leukoencephalopathy*	1/103 (0.9%)	
*Pulmonary and extrapulmonary TB MDR*	1/103 (0.9%)	
*CMV chorioretinitis*	2/103 (2.9%)	
*Progressive multifocal leukoencephalopathy*	1/103 (0.9%)	
*Extrapulmonary cryptococcosis*	1/103 (0.9%)	
*Esophageal candidiasis*	1/103 (0.9%)	
*Abdominal tubercolosis*	1/103 (0.9%)	
*Disseminated tubercolosis*	1/103 (0.9%)	
*Tuberculous Lymphadenitis*	1/103 (0.9%)	
*Chronic diarrhea*	1/103 (0.9%)	
*Thrombocytopenia*	1/103 (0.9%)	
*Cerebellartoxoplasmosi*	1/103 (0.9%)	
*Criptosporidium*	1/103 (0.9%)	
*Psoriasiform dermatoses*	1/103 (0.9%)	
*Encephalitis*	1/103 (0.9%)	
*CMV encephalitis*	1/103 (0.9%)	

**Table 2 viruses-14-02544-t002:** Primers employed in qPCR to detect and quantify *JCPyV*, *BKPyV*, *MCPyV*, *HPyV6* and *HPyV7 DNA*.

*HPyVs*	*Target*	*Sense*	*Antisense*	*Probe*	*References*
JCPyV	LT	5’-GAGTGTTGGGA TCCTGTGTTTTC-3’	5’-GAGAAGTGGGATGA AGACCTGTTT-3’	5′-FAM-TCATCACTGGCAAAC ATTTCTTCATGGC-MGB-3′	[33,34,35]
BKPyV	VP1	5′-AGTGGATGG GCA GCCTATGTA-3′	5′-TCATATCTGGGTCCC CTGGA-3′	5′VIC-TATGGAATCCCA GGTAGAAGA-MGB-3′	[34,36]
MCPyV	VP1	5′-TGCCTCCCACATCT GCAAT-3′	5′-GTGTCTCTGCCAA TGCTAAATGA-3′	5′-FAM-TGTCACAGGTAA TATC-MGB-3′	[37]
HPyV6	VP1	5′-GGCCTGGAAGGGC CTAGTAA-3′	5′-ATTGGCAGCTGTAAC TTGTTTTCTG-3′	5′-JOE-AGAACCAACCAT CTG TTG-BHQ1-3′	[38]
HPyV7	VP1	5′-AGGTCAATGAAGCC CTAGAAGGT-3′	5′-TGCTTTCTGAGGGC TTGCA-3′	5′-FAM-CAGGCAATACTG ATGTAGC-MGB-3′	[38]

**Table 3 viruses-14-02544-t003:** Primers employed in qualitative PCR to amplify NCCR, VP1 and MCPyV *LTAg*.

*HPyVs*	*Target*	*Sense*	*Antisense*	*References*
JCPyV	NCCR	**I N**: 5′-AGGGTCGAGCTCCATCATGG ATTCTTCC-3′	**I N**: 5′-CATGGTCCCCCAAAAGTGC TAGAGCAGC-3′	[40]
		**II N**: 5′-CCTCCACGCCCTTACTACTTCTGAG-3′	**II N**: 5′-AGCCTGGTGACAAGCCAAAACAGCTCT-3′	
	VP1	**I N**: 5′-CAATCTCAAGTCATGAACAC-3′	**I N**: 5′-GTCAACGTATCTCATCATGT-3′	[43]
		**II N**: 5′-TTTTGGGAGACTAACAGGAG-3′	**II N**: 5′-TAAAGCCTCCCCCCCAA CAGAAA-3	
BKPyV	NCCR	**I N**: 5′-AGGGTCGAGCTCCATCATGGA TTCTTCC-3′	**I N**: 5′-CATGGTCCCCCAAAAGTG CTAGAGCAGC-3′	[40]
		**II N**: 5′-GGCCTCAGAAAAAGCCTCCA CACCCTTACTACTTGA-3′	**II N**: 5′-CTTGTCGTGACAGCTGG CGCAGAAC-3′	
	VP1	**I N**: 5′-ATCAAAGAACTGCTCCTCAAA-3′	**I N**: 5′-GCACTCCCTGCATTTCCAAGGG-3′	[44]
		**II N**: 5′-CAAGTCCCAAAACTACTAAT-3′	**II N**: 5′-TGCATGAAGGTTAAGCATGC-3′	
MCPyV	NCCR	**I N**: 5′-AGAGAG CCTATACCACTAACAG-3′	**I N**: 5′-ACATGATTGAACTTTTATTG-3′	[41]
		**II N**: 5′-AATTTCACCAATATTGGCCAGCAG3′	**II N**: 5′-GAGGCGGAGTTTGACTGAT3′	
	VP1	**I N**: 5′-TGCAAATCCAGAGGTTCTCC-3′	**I N**: 5′-AAAACACCCAAAAGGCAATG-3	[41]
		**II N**: 5′-ATATTGCCTCCCACATCTGC-3′	**II N**: 5′-TGCCCTAATGTTGCCTCAGT-3′	
	LT	(**LT1**): 5’-TACAAGCACTCCACCAAAGC-3’	5’-TCCAATTACAGCTGGCCTCT-3’	[6]
	LT	(**LT3**): 5’-TTGTCTCGCCAGCATTGTAG-3’	5’-ATATAGGGGCCTCGTCAACC-3’	[6]

N: nested.

**Table 4 viruses-14-02544-t004:** HPyVs prevalence and mean viral load in urine and plasma of HIV-1-infected patients.

	*Urine*		*Plasma*	
	*n (%)*	*Viral Load (gEq/mL)*	*n (%)*	*Viral Load (gEq/mL)*
** *JCPyV* **	57/103 (55%)	6 × 10^7^	16/103 (15%)	6 × 10^5^
** *BKPyV* **	26/103 (25%)	1 × 10^5^	4/103 (3.8%)	2.5 × 10^3^
** *MCPyV* **	22/103 (21%)	5 × 10^3^	8/103 (7.7%)	1 × 10^3^
** *HPyV6* **	0/103		0/103	-
** *HPyV7* **	0/103		0/103	-
	** *Qualitative Detection* **			-
** *QPyV* **	0/103		0/103	-

n: number of patients; gEq/mL: genome Equivalents/milliLiter.

**Table 5 viruses-14-02544-t005:** HPyVs co-infections.

*HPyVs Co-Infection*	*Urine n (%)*	*Plasma n (%)*
JCPyV-BKPyV-MCPyV	12/103 (11.6%)	-
BKPyV-MCPyV	2/103 (1.9%)	-
JCPyV-MCPyV	7/103 (6.8%)	1/103 (0.97%)
JCPyV-BKPyV	9/103 (8.7%)	1/103 (0.97%)

n: number of samples.

**Table 6 viruses-14-02544-t006:** Analysis of HPyVs detection in relation to CD4+ cell count.

		*CD4+/mm^3^*		*p Value*
	*≤200 (n = 68)*	*>200 (n = 35)*	*Mean Value*	
**JCPyV + n**,%	40/68(58.8%)	19/35 (54.3%)	168.4	<0.05
**BKPyV + n**,%	17/68 (25%)	10/35 (28.6%)	216.8	<0.05
**MCPyV + n**,%	20/68 (29.4%)	12/35 (34.3%)	224.8	<0.05

**Table 7 viruses-14-02544-t007:** Sequence analysis of MCPyV NCCR.

*Sample*	*MCPyV + n*	*Archetype NCCR n (%)*
Urine	22/ 22 (100%)	22/22 (100%)
Plasma	8/8 (100%)	8/8 (100%)

n: number of patients.

**Table 8 viruses-14-02544-t008:** Sequence analysis of JCPyV NCCR.

*Sample*	*JCPyV + n*	*Archetype NCCR n (%)*	*Nucleotide Changes **		
			*B_37_T→G*	*C_108_G→A*	*F_217_G→A*
Urine	57	57/57 (100%)	-	-	-
Plasma	16	16/16 (100%)	11/16	2/16	9/16

n: number of patients; *: nucleotide changes.

**Table 9 viruses-14-02544-t009:** Sequence analysis of BKPyV NCCR.

*Sample*	*BKPyV + n*	*Archetype NCCR n (%)*	*Nucleotide Changes **	
			*R_5_A→G*	*P_19_A→G*
Urine	26	26/26 (100%)	-	-
Plasma	4	4/4 (100%)	3/4	1/4

n: number of patients; *: nucleotide changes.

## Data Availability

Data are contained within the article.

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
