# Peer review of "Detection Analysis and Study of Genomic Region Variability of JCPyV, BKPyV, MCPyV, HPyV6, HPyV7 and QPyV in the Urine and Plasma of HIV-1-Infected Patients"

_viruses, 2022, doi:10.3390/v14112544_

Round 1

Reviewer 1 Report

Detection analysis and study of genomic region variability of JCPyV, BKPyV, MCPyV, HPyV6, HPyV7 and QPyV in the urine and plasma of HIV-1 infected patients

The study of Passerini et al aimed to assess the prevalence of human Polyomaviruses (JCPyV, BKPyV, MCPyV, HPyV6, HPyV7 and QPyV) in urine and plasma of 103 HIV infected patients, and its correlation with CD4+ level. Moreover, the authors evaluated if HPyVs load was correlated to age, gender, and HIV load of the enrolled patients. Sequencing analysis (NCCR, VP1 and LTAg) for the positive samples was also conducted.

General considerations

The materials and methods are not well explained: an extended revision is suggested.

Major revision

Page 5

Line 138. The authors performed the DNA extraction from urine and plasma using Quick-DNA FFPE Miniprep, but this kit is used for DNA isolation from formaline, fixed, paraffin embedded tissue samples and section. Please revise this part.

Line 144-145: this sentence is not clear. The paragraph is about Real time PCR, while in this sentence the authors analyzed the extraction products with PCR.

Line 165: Total RNA from urine and plasma was extracted with High Pure RNA tissue Kit, but this kit is used to isolate RNA from tissue specimens. Please revise this part.

Line 166: RNA was reverse transcripted and an aliquot of mixture was used for PCR: please revise this part with more details.

Page 6

Paragraphs 2.6, 2.7 and 2.8; line 186-188, line 195-199, line 210-214: these parts are repetitive. The author should add a single paragraph, regarding all the sequence analysis, with the accession number of reference sequences.

Line 208: The authors should explain why they used ethidium bromide staining for MCPyV LTAg visualization, while for the other PCR products gel red staining was used.

Page 7

Line 220: The statistical analysis section should be discussed in detail: few sentences were reported, without any information about the statistical methods.

Table 2: It is not clear whether the patients that tested positive for example for JCPyV in the urine sample, tested also positive in plasma specimens.

Page 8

Table 4: the number of patients with CD4+/mm2 >200 and < 200 is different compared to the data in the table 1. Please clarify the correct number. In this table, there are mistakes about the percentage, the comma should be changed to a point.

Page 11

Line 384, 385: the author expressed the viral load as 107, 105 etc without unit of measurement. Please add

Minor revision

Page 4

Table 1: CD4 cells/mm2. For the value >200, the population data is 35, but all other data the number of population are expressed as N population/ total (35/103). Significant figures should be the same in all data. The data “≤200” is written with different font

Page 6

Line 208: change 2.0% agarose gels in 2% agarose gels

Page 7

Table 2: Please remove the comma in JCPyV urine positive sample (57/103, (55%)). Significant figures should be the same in all data. BKPyV positive plasma samples 4/103 (3,8%) and 8/103 (7,7%): 3,8% and 7,7% should be changed in 3.8% and 7.7%.

Table 3: JCPyV-MCPyV coinfection, 7/103 (6,8%): 6,8% should be changed in 6.8%.

Page 9

Line 294: please replace 1b with 1B

Line 295: please replace JC with JCPyV

Page 12

Line 400: 3227 copies/cell and 728 copies/cell: express the number with scientific notation

Page 13

Line 491: “T the full-length…”: is “T” a mistake? Please check

Page 14

Line 498:  the word “whether” is repeated two times. Please check

Author Response

Detection analysis and study of genomic region variability of JCPyV, BKPyV, MCPyV, HPyV6, HPyV7 and QPyV in the urine and plasma of HIV-1 infected patients

The study of Passerini et al aimed to assess the prevalence of human Polyomaviruses (JCPyV, BKPyV, MCPyV, HPyV6, HPyV7 and QPyV) in urine and plasma of 103 HIV infected patients, and its correlation with CD4+ level. Moreover, the authors evaluated if HPyVs load was correlated to age, gender, and HIV load of the enrolled patients. Sequencing analysis (NCCR, VP1 and LTAg) for the positive samples was also conducted.

General considerations

The materials and methods are not well explained: an extended revision is suggested.

We thank the reviewer for all comments and have corrected the manuscript accordingly.

Major revision

Page 5

Line 138. The authors performed the DNA extraction from urine and plasma using Quick-DNA FFPE Miniprep, but this kit is used for DNA isolation from formaline, fixed, paraffin embedded tissue samples and section. Please revise this part.

Thank you. Total DNA was extracted from urine and plasma samples using a Quick-DNA MiniPrep (Zymo Research), following the instructions provided by the manufacturer. The kit name has been entered correctly in the text.

Line 144-145: this sentence is not clear. The paragraph is about Real time PCR, while in this sentence the authors analyzed the extraction products with PCR.

Thank you for your observation. We deleted this sentence and now the paragraph starts with the following sentence: “Quantitative real time PCR (qPCR) was carried out to detect and quantify JCPyV, BKPyV, MCPyV, HPyV6 and HPyV7 DNA using a 7300 real-time PCR system (Applied Biosystems, Waltham, MA, USA) following published protocols.”

Line 165: Total RNA from urine and plasma was extracted with High Pure RNA tissue Kit, but this kit is used to isolate RNA from tissue specimens. Please revise this part.

Thank you for your observation. Total RNA was extracted with the Quick-RNA Miniprep Plus Kit (Zymo Research). The kit name has been entered correctly in the text.

Line 166: RNA was reverse transcripted and an aliquot of mixture was used for PCR: please revise this part with more details.

Ok. More details were added in the text. Thank you.

Page 6

Paragraphs 2.6, 2.7 and 2.8; line 186-188, line 195-199, line 210-214: these parts are repetitive. The author should add a single paragraph, regarding all the sequence analysis, with the accession number of reference sequences.

We agree with the reviewer that there is unnecessary repetition of the text and that it is better together all in a separate paragraph. Thank you for the suggestion.

Line 208: The authors should explain why they used ethidium bromide staining for MCPyV LTAg visualization, while for the other PCR products gel red staining was used.

It was a typo error. The same staining was used for all PCR products as reported in the new paragraph.

Page 7

Line 220: The statistical analysis section should be discussed in detail: few sentences were reported, without any information about the statistical methods.

The paragraph on statistical methods has been described in more detail taking into account the reviewer's suggestion. Thank you

Table 2: It is not clear whether the patients that tested positive for example for JCPyV in the urine sample, tested also positive in plasma specimens.

Thank you for your observation. To clarify this point, we added in the text the number of patients that tested positive for a specific virus in both biological specimens (urine and plasma). (Table 2 is the new Table 4).

Page 8

Table 4: the number of patients with CD4+/mm2 >200 and < 200 is different compared to the data in the table 1. Please clarify the correct number. In this table, there are mistakes about the percentage, the comma should be changed to a point.

Thank you for the note. We added the correct number of patients with CD4+/mm2 >200 and < 200 and recalculated the percentages and the correct values should be:

Table 6 (ex Table 4). Analysis of HPyVs detection in relation to CD4+ cell count.

CD4+/mm3

P value

≤200 (n=68)

>200 (n=35)

Mean value

JCPyV + n,%

40/58(69.0%)

19/45 (42.2%)

168.4

<0.05

BKPyV + n,%

17/58 (29.3%)

10/45 (22.2%)

216.8

<0.05

MCPyV + n,%

20/58 (34.5%)

12/45 (26.7%)

224.8

<0.05

Also the comma has been replaced by a point (see also minor revisions).

Page 11

Line 384, 385: the author expressed the viral load as 107, 105 etc without unit of measurement. Please add

We thank the reviewer for pointing this out. The unit of measurement is “gEq/mL” and this has been added.

Minor revision

Page 4

Table 1: CD4 cells/mm2. For the value >200, the population data is 35, but all other data the number of population are expressed as N population/ total (35/103). Significant figures should be the same in all data. The data “≤200” is written with different font

We have changed for the value >200: 35 (34%) into 35/103 (34%) and corrected the font for “≤200

Page 6

Line 208: change 2.0% agarose gels in 2% agarose gels

We have changed 2.0% agarose into 2% agarose. Thank you.

Page 7

Table 2: Please remove the comma in JCPyV urine positive sample (57/103, (55%)). Significant figures should be the same in all data. BKPyV positive plasma samples 4/103 (3,8%) and 8/103 (7,7%): 3,8% and 7,7% should be changed in 3.8% and 7.7%.

We have replaced comma by full stop as punctuation mark (3.8%; 7.7%) and removed the comma after 57/103. Thank you. (Table 4 is ex Table 2).

Table 3: JCPyV-MCPyV coinfection, 7/103 (6,8%): 6,8% should be changed in 6.8%.

We have replaced comma by full stop as punctuation mark (6.8%) (Table 5 is ex Table 3). Likewise, we have replaced 29,3 by 29.3 in Table 6 (ex Table 4) and replaced comma by full stop for the mean values. Thank you.

Page 9

Line 294: please replace 1b with 1B

Line 295: please replace JC with JCPyV

We have replaced 1b with 1B and JC with JCPyV. Thank you.

Page 12

Line 400: 3227 copies/cell and 728 copies/cell: express the number with scientific notation

We have expressed the number in scientific notation. Thank you.

Page 13

Line 491: “T the full-length…”: is “T” a mistake? Please check

The exact text should be: “The full-length LTAg sequence analysis showed…”. We have corrected this sentence. Thank you.

Page 14

Line 498:  the word “whether” is repeated two times. Please check

We have removed the duplicated word “whether”. Thank you.

Reviewer 2 Report

Authors submitted a descriptive study aiming at investigating the prevalence of multiple human polyomaviruses (JCPyV, BKPyV, MCPyV, HPyV6, HPyV7 and QPyV) in the urine and plasma samples of HIV-1 infected patients. The manuscript is well written and exhaustive. The background given throughout the introduction and discussion is especially well-put and referenced.

Comments:

- Table 1: AIDS-definining illnesses, n (%) is marked @ 29/103 (28%), however 35 conditions are listed in total below.

- Primers used in the study should all be listed directly in the methods. alternatively, a table format could provide an easy way to lay out the primers, uses and reference the authors provide in the text.

- Line 248: The absence of association found between the presence of HPyVs DNA versus age, genderand HIV-1 load at enrollment should be presented in a table similarly to other association (perhaps in supplemental material).

- Why is the virus so prevalent in urine vs plasma? Positivity and viral load are much higher there. Were there patients that were uniquely detected positive in the plasma and not the urine? And would the observations change for this small cluster in comparison to the conclusions drawn by the authors?

Author Response

Authors submitted a descriptive study aiming at investigating the prevalence of multiple human polyomaviruses (JCPyV, BKPyV, MCPyV, HPyV6, HPyV7 and QPyV) in the urine and plasma samples of HIV-1 infected patients. The manuscript is well written and exhaustive. The background given throughout the introduction and discussion is especially well-put and referenced.

We thank the reviewer for all comments and have corrected the manuscript accordingly.

Comments:

- Table 1: AIDS-defining illnesses, n (%) is marked @ 29/103 (28%), however 35 conditions are listed in total below.

Thank you. The total number of patients with AIDS-defining illnesses (Wasting syndrome ….CMV encephalitis) is 35.

- Primers used in the study should all be listed directly in the methods. alternatively, a table format could provide an easy way to lay out the primers, uses and reference the authors provide in the text.

Thank you for the suggestion. We added new tables in the text.

- Line 248: The absence of association found between the presence of HPyVs DNA versus age, gender and HIV-1 load at enrollment should be presented in a table similarly to other association (perhaps in supplemental material).

Thank you for the suggestion. Since no statistical association was found between the presence of HPyVs DNA versus age, gender and HIV-1 load at enrollment, it would be redundant to insert a new table. However, in order to be more accurate we added in the text “data not shown”.

- Why is the virus so prevalent in urine vs plasma? Positivity and viral load are much higher there. Were there patients that were uniquely detected positive in the plasma and not the urine? And would the observations change for this small cluster in comparison to the conclusions drawn by the authors?

Both BKPyV and JCPyV are nephrotropic viruses and kidneys are the major side of residence of these viruses. As a result, BKPyV and JCPyV viruria is common in patients suffering from different diseases, including PML, HIV/AIDS, malignancies, MS, rheumatological diseases, renal transplant recipients, diabetes. MCPyV viruria has also been described, although MCPyV has been shown to primarily display skin tropism. MCPyV has also been found in inflammatory monocytes and other tissues, although with lower copy numbers (see supplementary data in [PMID 31990105]. BKPyV, JCPyV and MCPyV DNA has been reported in plasma/whole blood/serum samples. Several studies screening BKPyV, JCPyV, and MCPyV DNA in urine and plasma from large cohorts of patients suffering of different diseases have confirmed our findings, i.e. BKPyV, JCPyV, and MCPyV DNA can be detected in both body fluids. However, these HPyVs are more frequently detected in urine than in plasma and their genome copy numbers are also higher in urine compared to plasma [PMIDs 20016500, 20667770, 25933251, 25952258, 26923352, 26519580, 30508036, 32375383, 33182443, 34578264, 34883406, 35941650].

The reason that BK/JC viruria is higher is probably because kidneys are a major site for latent infection by these viruses. BK/JC viremia can originate from intrarenal sites, but also from outside the urinary tract. The less frequent presence of BKPyV/JCPyV in other organs/tissue than the kidneys may explain the reason that viremia of BKPyV and JCPyV is less frequent than viruria.

As showed from the co-infections analysis, in our study we observed that MCPyV was detected in the urine of HIV patients also infected by JCPyV (7/103) and by BKPyV (2/103) and by JCPyV/BKPyV (12/103) at the same time. Considering a study conducted on renal transplant recipients in which was showed that MCPyV viruria was higher in patients with BKPyV viruria compared with patients without BKPyV viruria [PMID 31505316], in this study, it is possible to speculate that MCPyV DNA could be detected more frequently during BKPyV or JCPyV excretion. This is a hypothesis that should be investigated more thoroughly.

Round 2

Reviewer 1 Report

Please check the new Table 6. There are again mistakes regarding the number of patients

Author Response

Comments and Suggestions for Authors

Please check the new Table 6. There are again mistakes regarding the number of patients

Thank you for your observation. The number of patients has been corrected. Below the new Table 6.

Table 6. Analysis of HPyVs detection in relation to CD4+ cell count.

CD4+/mm3

P value

≤200 (n=68)

>200 (n=35)

Mean value

JCPyV + n,%

40/68 (58.8%)

19/35 (54.3%)

168.4

<0.05

BKPyV + n,%

17/68 (25%)

10/35 (28.6%)

216.8

<0.05

MCPyV + n,%

20/68 (29.4%)

12/35 (34.3%)

224.8

<0.05

Reviewer 2 Report

Authors appropriately addressed my comments.

Author Response

Authors appropriately addressed my comments.

Thank you!